# The Antimicrobial Resistance Index and Fournier Gangrene Severity Index of Patients Diagnosed with Fournier’s Gangrene in a Tertiary Hospital in North Eastern Romania

**DOI:** 10.3390/medicina59040643

**Published:** 2023-03-24

**Authors:** Dragoş Puia, Ştefan Gheorghincă, Cătălin Pricop

**Affiliations:** 1“Grigore T. Popa” Department of Urology, University of Medicine and Pharmacy, 700115 Iași, Romania; 2“C. I. Parhon” Hospital, Department of Urology, 700503 Iași, Romania; 3Neamţ Emergency County Hospital, 610136 Piatra Neamt, Romania

**Keywords:** Fournier’s gangrene, MAR-index, FGSI, antibiotic resistance, risk factor

## Abstract

*Background*: Although rare, Fournier’s gangrene is a major urological emergency. We aimed to learn more about the pathogenesis of Fournier’s gangrene and assess the antibiotic resistance patterns in individuals with this disease. *Methods*: We retrospectively evaluated the patients diagnosed with and treated for Fournier’s gangrene in a Neamt county hospital and “CI Parhon” Clinical Hospital in Iasi, Romania between 1 January 2016 and 1 June 2022. *Results*: We included a total of 40 patients, all males; of these, 12.5% died. In our study, in the patients that died, the adverse prognostic factors were a higher body temperature (38.12 ± 0.68 vs. 38.94 ± 0.85 °C; *p* = 0.009), an elevated WBC (17.4 ± 5.46 vs. 25.23 ± 7.48; *p* = 0.003), obesity (14.28% vs. 60%; *p* = 0.04), and a significantly higher FGSI (4.17 ± 2.80 vs. 9.4 ± 3.2; *p* = 0.0002) as well as MAR index (0.37 ± 0.29 vs. 0.59 ± 0.24; *p* = 0.036). These patients were more likely to have liver affections than those in the group who survived, but the difference was not significant. The most frequently identified microorganism in the tissue secretions culture was *E. coli* (40%), followed by Klebsiella pneumoniae (30%) and Enterococcus (10%). The highest MAR index was encountered in Acinetobacter (1), in a patient that did not survive, followed by Pseudomonas (0.85) and Proteus (0.75). *Conclusions*: Fournier’s gangrene remains a fatal condition, a highly resistant causative microorganism that is not always correlated with a poor prognosis.

## 1. Introduction

Fournier’s gangrene is a severe infection characterized by necrosis of the perineal teguments. Due to the severity and rapidity of its spread, Fournier’s gangrene is a major urological emergency. Its high mortality, usually up to 40% of cases according to Wróblewska et al., represents one of the main reasons why it is an important medical–surgical emergency and one of the main reasons infection causes blood vessel thrombosis, which leads to ischemia and necrosis of nearby soft tissue and fascia. The infectious and inflammatory process also spreads along the dartos fascia, Colle’s fascia, and Scarpa’s fascia, allowing the abdominal wall to be involved. Clinicians may miss this condition because the overlaying soft tissue is frequently inconspicuous due to the first fascial and subcutaneous involvement [1,2]. The treatment is based on hydro-electrolyte resuscitation, the vigorous excision of necrotic tissues, and parenteral broad-spectrum antibiotics. The initial antibiotic therapy should be based on a thorough understanding of the causative bacteria’s epidemiology and drug resistance pattern [2,3]. However, there are not many studies on the bacteriology of Fournier’s gangrene and the sensitivity patterns of the causative germs. As a result, our study aims to learn more about the pathogenesis of Fournier’s gangrene and assess the antibiotic resistance patterns in individuals with this disease; additionally, we want to highlight the factors that can predict a patient’s evolution. By identifying the local resistance pattern, we intend to develop antibiotic therapy schemes that will be applied until we have the result of the antibiogram.

## 2. Materials and Methods

We retrospectively evaluated the patients diagnosed with and treated for Fournier’s gangrene in Neamt county hospital and a tertiary center in the northeastern “CI Parhon” Clinical Hospital in Iasi, Romania between 1 January 2016 and 1 June 2022. Patients with no signs of skin and/or soft tissue necrosis and those with a single perianal, scrotal, or periurethral abscess were excluded from the study. We collected data regarding the patient’s age, comorbidities (e.g., arterial hypertension, diabetes mellitus, obesity (BMI > 30), or chronic kidney disease—CKD (for which we adopted the Kidney Disease Quality Outcome Initiative definition, according to which a glomerular filtration rate of <60 mL/min/1.73 m^2^ for 3 months or more, irrespective of cause, represents kidney damage), tissue culture, and antibiotic susceptibility of the identified pathogens. We also calculated Laor’s Fournier’s gangrene severity index (FGSI) for each patient using the following variables recorded at admission: temperature, heart rate, respiration rate, and the serum levels of sodium, potassium, creatinine, and bi-carbonate concentrations, hematocrit, and white blood counts (WBCs) [4]. The deviation from normal values was graded from 0 to 4 on a scale of 0 to 4. Pathologic fluids from the necrotic tissue were inoculated on agar plates. Antibiogram was obtained using the Kirby–Bauer disk diffusion test. Krumperman’s formula was used to calculate the multiple antibiotic resistance (MAR = a/b) index, where “a” was the number of antibiotics against which the test isolate showed resistance and “b” was the total number of antibiotics against which the test isolate was assessed for susceptibility [5]. The statistical analysis was performed using the Student’s *t*, ANOVA, and Fisher tests. We considered a *p*-value of < 0.05 to be statistically significant. The correlation coefficient was calculated using Spearman’s Rho test. We applied the tests online by using the www.socscistatistics.com website. A logistic regression model was also applied using an online calculator available online at https://stats.blue/Stats_Suite/logistic_regression_calculator.html, accessed on 16 November 2022. We included a total of 40 patients, all males, that were aged between 39 and 90 years old. In all of the cases, the extensive debridement of necrotic tissue and empiric broad-spectrum antibiotic therapy (with at least two drugs/molecules) were performed in the first 6 h from admission. Because some types of complementary surgical therapies, such as hyperbaric oxygen therapy and negative pressure wound therapy, were not available, the surgical wounds were standardly treated. 

## 3. Results

We included a total of 40 patients, all males, that were aged between 39 and 90 years old. In all of the cases, the extensive debridement of necrotic tissue and empiric broad-spectrum antibiotic therapy (with at least two drugs/molecules) were performed in the first 6 h from admission. Because some types of complementary surgical therapies, such as hyperbaric oxygen therapy and negative pressure wound therapy, were not available, the surgical wounds were standardly treated. Out of the 40 patients, five deceased, resulting in a total mortality of 12.5%. In three of these patients, one death occurred in the first 24 h after hospitalization and the other two occurred on days 19 and 26 respectively. As shown in Table 1, the lengths of hospital stays for these five patients were significantly shorter compared to those of the other patients. In our study, in the patients who died, the adverse prognostic factors were a higher body temperature, an elevated WBC, obesity, and a significantly higher FGSI as well as MAR index, as shown in Table 1. These patients were more likely to have liver affections than those in the group who survived, but the difference was not significant.

The most frequently identified microorganism in the tissue secretions culture was *E. coli* (40%) (*n* = 16), followed by *Klebsiella pneumoniae* (30%) (*n* = 12) and *Enterococcus* (10%). The highest MAR index was encountered in *Acinetobacter*, in a patient that did not survive, followed by *Pseudomonas*, as shown in Table 2. Regarding the sensitivity patterns in the most encountered microorganisms, *E. coli* had an 81.25% sensitivity to carbapenems, 68.75% to piperacillin–tazobactam, 50% to ceftazidime, and 43.75% to levofloxacin, while *Klebsiella* showed an overall sensitivity of 83.33% to carbapenems, 58.33% to piperacillin–tazobactam, 50% to ceftazidime, and 41.66% to levofloxacin as well as ceftriaxone.

The correlation between the FGSI and the MAR index was weak (r = 0.206), which was also the case between the WBC and the MAR index (r = 0.02). We found a negative correlation in deceased patients between the FGSI and the MAR index (r = −0.65). The correlation between the MAR index and mortality was also weak (r = 0.03727, *p* = 0.924). Previous urethral catheter indwelling was noted in four patients, all of whom were in the group of those who survived. In our study, 72.5% (n = 29) of the patients received three types of antibiotics; all deceased patients were treated with three antibiotics from admission. As seen in Table 3, Metronidazole was used in 90% (n = 36) of the cases and in all of the deceased patients. Other antibiotics used included carbapenems (50%) (n = 20), vancomycin (37.5%) (n = 15), fluoroquinolones (20%) (n = 8), cephalosporins (17.5%) (n = 7), piperacillin and tazobactam 12.5% (n = 5), and gentamicin 5% (n = 2). The most frequently used combination was vancomycin plus metronidazole and carbapenem, used in 37.5% (n = 15) cases.

We also arbitrarily divided the patients according to their hospitalization periods. We compared the outcomes, mean MAR indexes, FGSIs, and treatment regimens. The results are shown in Table 3.

## 4. Discussion

Fournier’s gangrene is a major urological emergency due to its aggression and death rate. According to Vick et al., all of the cases in Fournier’s original version were reported as being idiopathic. In the examination of a more recent series, a cause was identified in 75% to 100% of the cases. Thirteen percent to fifty percent of these cases had colorectal origins. Seventeen percent to eighty-seven percent of the sources were found to be urogenital [2]. The two remaining causes were localized trauma and skin infections. The more frequently documented colorectal causes are colonic perforations brought on by malignancy, trauma, and diverticulitis, as well as perirectal, perianal, and ischiorectal abscesses, in addition to rectal instrumentation. Balanitis, urethral instrumentation, and urethral strictures with urine extravasation are common urogenital sources. The infection typically begins as insidious cellulitis close to the portal of entry, frequently in the perineum or perineal region. The presence of anaerobes in the illness is believed to be responsible for the characteristic feculent odor of the affected area, which is frequently swollen, dark, and covered in macerated skin. Additionally, patients may experience strong systemic symptoms that are frequently unrelated to the local severity of the illness. The general health of patients with a severe clinical presentation declines as the gangrenous process progresses to mal-odorous discharge and sloughing in the afflicted locations. According to Wróblewska et al., the mortality rate is usually between 20 and 40%; some authors have reported a mortality of up to 88%, while in a very large series of 1680 patients, Sorensen et al. noticed an overall mortality rate of 7.5%, which is much smaller and similar to that of our group [1,3,6]. Fortunately, it is a condition rarely encountered in daily practice; according to Sorensen et al., it has an incidence of about 1.6/per 100,000 men, representing fewer than 0.02% of hospitalized patients [6].

Being a disease with a fulminant evolution, many scores have been created to assess its evolution. One of the most used is the FGSI, created in the mid-1990s by Laor et al. [5], which took into account the following parameters: body temperature, heart rate, respiratory rate, WBC, hematocrit, and the serum levels of sodium, potassium, creatinine, and bicarbonate. Each case ultimately underwent the measurement of these nine parameters, with the degree of variation from normal being evaluated from zero to four. The individual data were added up to calculate Fournier’s gangrene severity index score. The data were segmented based on the patient’s results, or whether they survived. Changes in the serum albumin, total protein, and cholesterol levels represented the severity of the dysfunction and the worse prognoses in their series. However, individually, these findings did not seem applicable to the clinical setting, and in the end, they were not included in the final version of the FGSI. According to the authors, a score greater than nine is highly correlated with mortality. In a group of 83 patients, Noegroho et al. recorded patients that survived with an average FGSI score of 5.5, which was higher than that in our case, while in those who had deceased the average score was 14 [7]. The FGSI was updated (UFGSI) in 2010 by Yilmazlar et al., who added two new parameters: age and disease severity [8]. Some authors have compared the two scoring systems. Roghmann et al. applied the scores to 44 patients and concluded that all scoring systems can be used to estimate mortality [9]. However, even though the UFGSI has more variables, it does not appear superior to the FGSI. The scoring systems are far from perfect. In a multicenter study, Üreyen et al. proposed the addition of the extension of the necrosis. At the same time, Çomçali et al. suggested that albumin and the need for positive inotropic support are independent risk factors for mortality [10,11].

Some authors dispute the usefulness of the FGSI score in favor of other elements. Kahramanca et al. compared the neutrophil–lymphocyte ratio and the platelet–lymphocyte ratio with the FGSI in determining prognosis in 64 patients [12]. The authors reported a significantly higher neutrophil–lymphocyte ratio and platelet–lymphocyte ratio (*p* < 0.001) in patients that required multiple debridement procedures, while there was no difference in the FGSI (*p* = 0.121). Elevated neutrophil–lymphocyte and platelet–lymphocyte ratios in non-survivor patients have also been noted by Yim et al.; however, in their study, the FGSI was also significantly higher in the deceased [13].

We note that none of the scores take into account the comorbidities of patients, of which diabetes mellitus is one of the most encountered; in our group, the incidence of it was 57.5%. In a study by Yilmazlar et al., it was the most common comorbidity in the patients, noticed in 64% of the cases. In contrast, in another study, by Lin et al., 87.7% of the deceased patients had this diagnosis [14,15]. Although diabetes is considered a common source of weakened immunological responses, and it might increase a patient’s susceptibility to sepsis, interestingly, as in our case, many authors have not noticed differences between groups from this point of view [9,14,15,16]. However, in a group of 34 patients, Şahin et al. noticed that significantly more patients had diabetes in the non-survivor group. Additionally, in a meta-analysis of 15 studies by El-Qushayri et al., diabetes was significantly more frequently detected in patients with higher mortality rates (RR 0.72, 95% CI 0.59–0.89; *p* < 0.01) [17,18]. Other elements associated with elevated mortality identified by authors have been heart disease, renal failure, and chronic kidney disease. These data are in contradiction with those reported in our study. In our study, only obesity was significantly more encountered in deceased patients; other authors, such as Eğin et al., did not find any significant difference between groups from the point of view of obesity (*p* = 0.86). However, there was an overall significant difference regarding comorbid conditions other than diabetes and obesity (*p* = 0.004) [19].

In our patients, we did not have individuals with known onco-hematological diseases. Although, this could be a neglected predisposing factor. In a recent systematic review by Creta et al., 35 studies, including 42 patients, were identified [20]. The youngest patient was only four years old. The most frequently diagnosed onco-hematological disease was acute myeloid leukemia in 50% of cases and acute lymphocytic leukemia in 21.42%. Interestingly, in 23.80% of cases, Fournier’s gangrene was the symptom of the malignancy. We could suppose that an impaired immunological system was an important predisposing factor in these patients because, at the moment of the diagnosis, 40.9% received chemotherapy while 9.1% had a recent stem cell transplant performed. As for the pathogens involved, the most frequently identified has been Pseudomonas aeruginosa in 61.5% of cases, followed by *Escherichia coli* [20].

The European Association of Urology classifies Fournier’s gangrene as a type 1 necrotizing fasciitis of a polymicrobial etiology. According to Wróblewska et al., enteric rods, Gram-positive cocci, and obligate anaerobic bacteria are common bacteria recovered from Fournier’s gangrene patients [1]. In our study, the most commonly identified bacteria were *E. coli*, followed by Klebsiella. *E. coli* were also the most commonly identified bacteria by Yilmazlar et al., although, in their patients, they were followed by *Enterococcus* sp. (62%) and Acinetobacter baumannii (30%) [14]. According to Lin et al., the most frequently isolated germ in Taiwanese patients was *Streptococcus* spp., followed by *Peptoniphilus* spp. and Staphylococcus aureus [15]. In rare cases, the etiological agent may be a fungus, as in the cases reported by Johnin or Temiz [21,22]. In addition, the authors reported a high antibiotic resistance: 66% of Gram-negative bacilli strains were resistant to amoxicillin–clavulanic acid, and 38.1% showed resistance to piperacillin. Piperacillin–tazobactam resistance was found in one isolate. Fluoroquinolone resistance was found in 23.8% of the strains, while three isolates were resistant to ceftriaxone. Chia et al. also reported high antibiotic resistance; in a group of 59 patients, the authors reported a 21% incidence of multidrug-resistant organisms, multidrug-resistant Staphylococcus aureus being the most frequently identified [23]. Additionally, in a cohort of 40 patients, Bjurlin et al. reported 13% resistance to fluoroquinolones and 40% to trimethoprim/sulfamethoxazole for *E. coli* isolates. *Providencia* spp., *Klebsiella* spp., *E. coli*, and methicillin-resistant S. aureus were resistant to ampicillin–sulbactam. More importantly, the authors did not encounter resistance to clindamycin, vancomycin, or piperacillin–tazobactam [24]. In a very large series of 143 patients, Castillejo Becerra et al. noted that the majority of isolated microorganisms were sensitive to ampicillin–sulbactam, ceftriaxone, piperacillin–tazobactam, amikacin and cefepime, and resistant to ampicillin, trimethoprim–sulfamethoxazole, levofloxacin, and clindamycin [25]. In onco-hematological patients, Creta et al. highlighted, as first-line empirical antibiotic treatment, aminoglycosides followed by cephalosporines, glycopeptides, and lincosamides. After obtaining the blood culture result, the antibiotic regimen was changed, and in this setting, the most frequently used are aminoglycosides, followed by cephalosporins, carbapenems, and polymyxins [20]. Also, in a case report by Dragomir et al. of a patient with Fournier’s gangrene secondary to an intrarectal foreign body, the culture from the secretions was positive for multidrug-resistant Klebsiellosis. The antibiotic sensitivity was noticed only for ciprofloxacin, imipenem, and tetracycline [26].

The causative microbes, either aerobes or anaerobes, work together to produce a variety of exotoxins and enzymes that aid in the breakdown of tissue and the spread of infection, such as collagenase, heparinase, streptokinase, hyaluronidase, and streptodornase. Microvascular thrombosis and cutaneous necrosis are brought on by the aerobes’ induction of platelet aggregation and complement fixation, as well as the anaerobes’ production of heparinase and collagenase. Additionally, the necrotic tissue decreases phagocytic activity, which contributes to the infection’s continued progression. According to Chawla et al., a very important therapeutic component is surgical debridement, which aims to eliminate all nonviable tissues, stop the spread of infection, and lessen systemic toxicity [27]. To prevent severe sloughing and infection, appropriate medical and surgical care is required [28].

Although technically not available to our patients, there are complementary methods to surgery. Hyperbaric oxygen therapy can be a viable complementary treatment. According to Lewis et al., taking into account the etiology of Fournier’s gangrene, the ischemia and necrosis brought on by arterial artery thrombosis create an ideal habitat for the growth of anaerobic bacteria. Therefore, bacterial growth is slowed down by creating an environment with sufficient oxygen [29]. Patients who are refractory to traditional therapies, such as sterile honey and maggots, should adopt this therapy approach in addition to early surgical debridement. However, because this treatment is usually offered to a more severe status, according to Schneidewind et al., there is an increased mortality rate in patients receiving this type of therapy, although some authors, such as Anheuser et al., reported 0% mortality in these patients. Another alternative after surgical debridement is negative pressure wound therapy [30,31], which is based on the idea that negative pressure vacuuming causes an increase in blood flow and the migration of inflammatory cells to the affected area. As a result, granulation tissue forms and bacterial contamination, toxins, exudates, and debris are cleared away. Ioannidis et al. encourage physiological healing. Michalczyk et al. reported 0% mortality in patients using a combination of hyperbaric oxygen therapy and negative pressure wound therapy compared with standard wound care [32,33].

In addition to fluid resuscitation, antibiotic therapy is essential to medical management. Blood, tissue, and urine cultures of aerobic, anaerobic, and fungal organisms should be obtained, after which antibiotic therapy should be started immediately. According to the European Urology Guidelines, there is no standard antibiotic regimen; the suggested regimens are piperacillin–tazobactam plus vancomycin, cefotaxime plus metronidazole or clindamycin, or carbapenem (imipenem, meropenem, or ertapenem) [34]. According to Huayllani et al., vancomycin or daptomycin, plus carbapenem or piperacillin–tazobactam, should be used as a first-line treatment [35]. In a recent survey by Schneidewind et al., the most often antibiotics combination used in Europe is metronidazole plus cephalosporine plus aminoglycoside, followed by piperacillin/tazobactam [36]. If there are concerns about toxin generation, clindamycin can be added. Local antibiograms should be analyzed to allow optimal coverage to be customized based on local medication resistance at that hospital/community.

According to Urbina et al., there are disparate practices regarding the duration of treatment. Stevens et al. suggest continuing medication for 48–72 h after debridement surgery, as this time frame appears sufficient to evaluate clinical progress, including the absence of fever [37,38]. Antibiotic de-escalation based on microbiological evidence from blood cultures and preoperative samples seems suitable, despite the lack of precise necrotizing soft tissue infection data [38].

The retrospective design and the small number of patients were the most significant limitations of our study. We also recognize that the mortality analysis may be insufficient due to the small number of patients. Furthermore, due to the impossibility of acquiring all data and the need for a uniform laboratory panel for each patient, a full analysis of the laboratory data could not be undertaken.

## 5. Conclusions

Although we have seen a lot of progress lately in diagnostic and treatment approaches, Fournier’s gangrene remains a fatal condition with high hospital costs. In our group, a higher body temperature, an elevated WBC, obesity, and a significantly higher FGSI and MAR index were negative prognostic factors. Still, diabetes was associated with only 57.5% of the cases. Knowing the parameters that predict hospitalization time and fatality rates enables patient-centered treatment and may help forecast more radical treatments or the need for extra treatment in high-risk patients.

## Figures and Tables

**Table 1 medicina-59-00643-t001:** Characteristics of included patients.

	Survivors	Deceased	Total	*p*
No. of patients	35	5	40	
Mean age (SD)	65.57 (±12.84)	66.8 (±6.11)	65.72 (±12.21)	0.75
Body temperature (°C)	38.12 (±0.68)	38.94 (±0.85)	38.22 (±0.74)	0.009
Hemoglobin (g/dL)	11.07 (±2.43)	10.61 (±1.33)	11.02 (±2.32)	0.34
WBC	17.4 (±5.46)	25.23 (±7.48)	18.38 (±6.28)	0.003
C reactive protein mg/L	277.38(±135.78)	192.62(±168.09)	265 (±139.67)	0.174
Platelet count (×10^3^/mm^3^)	183.78 (±26.56)	162.32 (±18.63)	181.45 (±25.32)	0.08
Diabetes mellitus (*n*)	57.14% (20)	60% (3)	57.5% (23)	0.9
Arterial hypertension (*n*)	57.14% (20)	60% (3)	57.5% (23)	0.9
Heart rate (bpm)	98.31 (±7.75)	106.6 (±10.4)	99.35 (±8.43)	0.019
CKD (*n*)	20% (7)	40% (2)	22.5% (9)	0.31
Hepatopathy (*n*)	31.42% (11)	20% (1)	30% (12)	0.98
Obesity (*n*)	14.28% (5)	60% (3)	20% (8)	0.04
BMI	24.47 (±5.21)	22.01 (±2,74)	24.17 (±5.01)	0.16
FGSI	4.17 (±2.80)	9.4 (±3.2)	4.8 (±3.31)	0.0002
Mean MAR index	0.37 (±0.29)	0.59 (±0.24)	0.4 (±0.29)	0.036
Length of stay (days)	19.48 (±7.99)	9.6 (±12.03)	18.25 (±9.03)	0.009

**Table 2 medicina-59-00643-t002:** Incidence of identified germs and the mean MAR index.

Identified Germ	Patients (*n*=)	Mean MAR Index (a/b)
*E. coli*	40% (16)	0.34
*Klebsiella*	30% (12)	0.39
*Enterococcus*	10% (4)	0.22
*Pseudomonas*	5% (2)	0.85
*Proteus mirabilis*	2.5% (1)	0.75
*S. aureus*	5% (2)	0.62
*Streptococcus* spp.	2.5% (1)	0.12
*Acinetobacter*	2.5% (1)	1
*Mixed flora*	2.5% (1)	0.56

**Table 3 medicina-59-00643-t003:** Differences between treatment regimens according to the length of stay.

	1–10 Days	11–20 Days	>20 Days
No of patients	7	17	16
Deceased (*n*)	3	1	1
Mean MAR index	0.32	0.32	0.50
Mean FGSI	7.14	4.35	4.31
3 antibiotics (*n*)	4	13	12
Metronidazole (*n*)	6	14	16
Carbapenems (*n*)	3	12	5
Vancomycin (*n*)	2	9	4
Fluoroquinolones(*n*)	1	1	6
Cephalosporines (*n*)	3	2	2
Piperacilin/Tazobactam (*n*)	1	3	1
Gentamicin (*n*)	1	1	0

## Data Availability

The datasets used and/or analyzed during the current study are available from the corresponding author on reasonable request.

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
