# Peer review of "The Antimicrobial Resistance Index and Fournier Gangrene Severity Index of Patients Diagnosed with Fournier’s Gangrene in a Tertiary Hospital in North Eastern Romania"

_medicina, 2023, doi:10.3390/medicina59040643_

Round 1
Reviewer 1 Report
I have made extensive comments on the article. The report involves a small sample size with a rare disease, FG. Table 1 and 2 could be merged. Hence, this could be better packaged as a Short communication.
1.The abstract needs extensive correction as the focus was only on the 12.5%.
2. The Introduction could not first highlight what FG is.
3. The tables in the Results were not drawn rightly. Only the horizontal lines are used in tables. The presentation of the results were not systematic. Some presented results do not have tables. Table 3 was not described.
I have attached the articles with corrected highlights and markups.

Author Response
Dear Reviewer, we have made correction in the manuscript file as you suggested. In addition here are some answers to you r questions:
Answer: we searched for the patients diagnosed and treated in that period, and as a RESULT, we identified 40 individuals in total
Answer: We did not focus on the 12.5%; we compared those who died with the survivors
..what do you mean by patients' evolution?. The work did not predict any.
Answer: Through patients’ evolution, we were referring to their survival. We highlighted these factors in the Conclusion
Your recommendation for extensive editing of English language and style confuse us, the manuscript was verified before uploading though the MDPI English proofing service. I have attached the certificate.

Reviewer 2 Report
1. Why is a wide range of tests (Student's t, ANOVA, and Fisher tests) used for statistical analysis? What was the purpose of logistic regression?
2. Why is it that after doing all these statistical tests, only a few cases of p-value are mentioned and the rest are reported based on percentages?
3. Why is Krumpermann's formula used to determine multiple antibiotic resistance, while WHO and CDC have proposed newer methods?
4. All bacteria names should be written in italics.
5. Why does this research not have an ethical code?
Author Response
1.Why is a wide range of tests (Student's t, ANOVA, and Fisher tests) used for statistical analysis? What was the purpose of logistic regression?
We analyzed many factors, and for this reason, we applied many tests (e.g. we could not apply the chi-square test because of the small number of deceased patients, in this case, Fisher test was used).
- Why is it that after doing all these statistical tests, only a few cases of p-value are mentioned and the rest are reported based on percentages?
P-values that are not mentioned in the text can be found in Table 1.
- Why is Krumpermann's formula used to determine multiple antibiotic resistance, while WHO and CDC have proposed newer methods?
Krumpermann's formula is straightforward and it is still widespread.
- All bacteria names should be written in italics.
We corrected this issue
- Why does this research not have an ethical code?
Dear reviewer, this is a retrospective observational study; we did not make any intervention (e.g. drugs, surgery) to obtain data for this study.
Reviewer 3 Report
The manuscript intends to study the pathogenesis of Fournier's gangrene and the antibiotic resistance associated with the microbes involved. This is a very relevant topic as this condition causes huge morbidity and mortality. It is missing majority of the details required for the readers to verify the results. Materials and method section lacks a lot of detail on the procedures that were done and this makes it difficult for the reviewer to evaluate the results. for example, Fournier's syndrome as stated in the discussion is also caused by anaerobic bacteria and fungus. So were there any methods used to identify these organisms. Also, the authors did not mention how data was collected, how was the sample size calculated, how bacteria were isolated, how the antimicrobial susceptibility testing was done, why only males were selected etc. Additionally, introduction does not provide sufficient rationale for the study and did not specify the specific objectives or the hypothesis. The Table presented in the results section does not provide sufficient details for the readers. Finally, the discussion section was more like a literature review and did not discuss the results obtained from this study. Certain references were missing in the text. There were no IRB statement. Overall, the manuscript is poorly written and needs a thorough re revision.
Author Response
Dear Reviewer, in our geographical area, there is a lack of information regarding local sensitivity patterns of urinary tract infections and less on Forunier’s gangrene. This is a retrospective observational study; we studied the medical record. Since no active intervention (e.g. drugs, surgery) for this study was performed, no IRB statement was obtained. If you notice from the affiliations, we are working in a urological tertiary center. FG is a necrotizing fasciitis of the perineal and genital superficial tissues. Women with FG are not referred to our clinic but to gynecology or general surgery. In addition, we added how antimicrobial susceptibility testing was done, and we several other modifications to the manuscript as suggested by the other reviewers.
Round 2
Reviewer 1 Report
The only correction required is the title where the abbreviation FG was used instead of written in full.
Author Response
Modified as requested
Reviewer 2 Report
Dear authors
1. Why does this research not have a code of ethics?
2. Why have they used various tests to perform a simple statistical analysis?
Author Response
- Why does this research not have a code of ethics?
Dear Reviewer, we could say that our study was conducted according to the general principles of the Declaration of Helsinki (ensure respect for all human subjects and protect their health and rights). However, this was not a prospective study. From this point of view, our patients were not at any risk. According to article 10 of the Declaration, “It is the duty of physicians involved in medical research to protect the life, health, dignity, integrity, right to self-determination, privacy, and confidentiality of personal information of research subjects.” Besides, the confidentiality of personal information, which was respected in our retrospective study, did not influence the patient’s treatment, dignity, integrity, etc. Also, although informed consent for the medical procedures (surgery, blood transfusion, etc.) was obtained for each patient, it was not the case for this research in particular. We did not administer experimental treatment, no randomization in different treatment groups, and but not least, we had no placebo group.
2. Why have they used various tests to perform a simple statistical analysis?
Besides the descriptive statistics, we used four tests:
Student t was used to compare quantitative data (e.g., body temperature, CRP, BMI, etc.) between the two groups (survivors and deceased patients). ANOVA was also used to compare quantitative data in more than two groups (e.g., as quantitative data from Table 3). Because in the group of deceased patients, we had only five persons, for the categorical data (e.g., no of patients with CKD or diabetes), we had to use the Fisher test; Chi-square could not be applicable. Finally, Spearman-Rho test highlighted any correlation between a specific factor and a patient’s outcome. In the end and there are not too many tests, and each has its justification.